# Anticoagulants and Osteoporosis

**DOI:** 10.3390/ijms20215275

**Published:** 2019-10-24

**Authors:** Salvatore Santo Signorelli, Salvatore Scuto, Elisa Marino, Michele Giusti, Anastasia Xourafa, Agostino Gaudio

**Affiliations:** Department of Clinical and Experimental Medicine, University of Catania, 95123 Catania, Italy; salvo.scuto1982@hotmail.it (S.S.); marinoelisa@msn.com (E.M.); michelegiusti88@gmail.com (M.G.); axourafa@gmail.com (A.X.); agostino.gaudio@gmail.com (A.G.)

**Keywords:** osteoporosis, fracture, anticoagulant agents, heparin, low molecular weight heparins (LMWHs), warfarin

## Abstract

Anticoagulant agents are widely used in the treatment of thromboembolic events and in stroke prevention. Data about their effects on bone tissue are in some cases limited or inconsistent (oral anti-vitamin K agents), and in others are sufficiently strong (heparins) to suggest caution in their use in subjects at risk of osteoporosis. This review analyses the effects of this group of drugs on bone metabolism, on bone mineral density, and on fragility fractures. A literature search strategy was developed by an experienced team of specialists by consulting the MEDLINE platform, including published papers and reviews updated to March 2019. Literature supports a detrimental effect of heparin on bone, with an increase in fracture rate. Low molecular weight heparins (LMWHs) seem to be safer than heparin. Although anti-vitamin K agents (VKAs) have a significant impact on bone metabolism, and in particular, on osteocalcin, data on bone mineral density (BMD) and fractures are contrasting. To date, the new direct oral anticoagulants (DOACs) are found to safe for bone health.

## 1. Introduction

For a long time two classes of anticoagulant drugs have been applied in the treatment of venous thromboembolism (VTE): heparin (and its derivatives) and oral anticoagulants [1]. Heparin was discovered in 1916 by Jerry McLean, a student working with professor W.H. Howell at John Hopkins University. Since 1937, heparin has been used in the prevention of pulmonary embolism, then also in acute VTE. Heparin concurs with the antithrombin III to accelerate the origin of a molecular complex including antithrombin plus serine proteases, thus inhibiting the activated coagulation factors II and X. Heparin shows fast, non-prolonged anticoagulant activity and low bioavailability, particularly at low dosage [2]. Low molecular weight heparins (LMWHs), originating from the depolymerisation of heparin, were introduced as anticoagulant drugs 10 years ago. LMWHs ensure therapeutic activity, showing high affinity with antithrombin III and inhibition of activated coagulation factors II and X. They also show reduced anti-platelet activity; consequently, these drugs have a low risk of inducing heparin-induced thrombocytopenia (HIT). Moreover, they can be used in pregnant women affected by VTE [3,4]. In contrast to heparin, LMWHs do not need coagulation-monitoring tests. All these characteristics make them more manageable than heparin in the treatment of VTE. Since the study performed by Brandajes [5], oral anti-vitamin K agents (VKAs) play a fundamental role in both the prevention and treatment of VTE. Most recently published guidelines for VTE indicate oral VKA as the first-line therapy for VTE after initial therapy with LMWHs [6,7,8,9,10]. VKAs produce their anticoagulant effect by interfering with the cyclic interconversion of vitamin K and its 2,3-epoxide, thereby modulating the γ-carboxylation of glutamate residues (Gla) on the *N*-terminal regions of vitamin K-dependent proteins [6]. The vitamin K-dependent coagulation factors II, VII, IX, and X require γ-carboxylation for their procoagulant activity, and treatment with VKAs results in the hepatic production of partially carboxylated and decarboxylated proteins with reduced coagulant activity. Warfarin and acenocumarol are the most widely used VKA drugs. Bleeding is the major adverse effect of VKAs; thus, a close monitoring of coagulation (International Normalised Ratio - INR test) is mandatory for patients assigned to VKAs. Consequently, bleeding risk is one of the most important limits and discomforts in long-term VKA treatments [11,12,13]. The numerous indications for VKA therapy [6,9,14,15,16] are reported in Table 1.

In recent years, four direct oral anticoagulants (DOACs): dabigatran, rivaroxaban, apixaban and edoxaban, have been approved for use in stroke prevention in nonvalvular atrial fibrillation (AF) and in the treatment of VTE. Dabigatran, a direct thrombin inhibitor, and rivaroxaban, apixaban and edoxaban, the factor Xa inhibitors, produce a more predictable, less labile anticoagulant effect; they have been shown to be at least as safe and effective as warfarin in stroke prevention in AF [17].

The aim of this review is to evaluate the impact of this group of drugs on bone. In particular, it analyses the effects of heparin, LMWHs, VKAs and DOACs on bone metabolism, bone mineral density, and fragility fractures.

## 2. Data Source and Search

A literature search strategy was developed by an experienced team of specialists by consulting the MEDLINE platform. The literature search performed including published papers and reviews updated to 2019. The search strategy used a combination of controlled key words (e.g., ‘Heparin’, ‘Low Molecular Weight Heparin’, ‘Vitamin K antagonist’, ‘DOAC’, ‘Osteoporosis’, ‘Bone metabolism’, ‘Fracture’). Search results were limited to papers published in English. Each participant in the search process extracted relevant information, and other participants verified the accuracy and completeness of the data. Each reviewer made a judgement whether the reported results from the search process were different from or corrected by findings from subsequent papers.

## 3. Heparin

The use of heparins is now widespread and consolidated. Unfractionated heparin (UFH) is an effective anticoagulant that is used principally for prophylaxis and the treatment of thromboembolic disorders and as an anticoagulant for extracorporeal circulation and dialysis procedures. UFH has a number of limitations and potential complications, including HIT, skin reactions, and osteoporosis with long-term use [18,19,20,21]. LMWHs were developed in the 1980s and many of the drawbacks related to heparin are overcome by their use, including a lower risk of osteoporosis [22,23,24].

### 3.1. Effects on Bone Metabolism

Several in vitro and animal studies on the effects of heparin on osteogenesis showed controversial results concerning osteogenic outcome. The mechanism behind osteoporosis is only partially understood.

Miur et al. have studied the effects of heparin on rat bones both histomorphometrically and biochemically. Histomorphometric analysis demonstrated that while both UFH and LMWH produced a dose-dependent decrease in cancellous bone volume, the effects of UFH were dramatically greater than those of LMWH. These studies have suggested that while both UFH and LMWHs decrease bone formation (by decreasing both osteoblast and osteoid surface), only UFH was found to increase bone resorption (by increasing osteoclast surface). These findings were supported by analysis of biochemical markers of bone turnover: Both UFH and LMWH treatment produce a dose-dependent decrease in serum alkaline phosphatase (a marker of bone formation), whereas only UFH causes a transient increase in urinary type I collagen cross-linked pyridinoline (PYD, a marker of bone resorption) [25,26].

In 1995 a study on calcium loss from foetal rat calvaria carry out by Shaughnessy et al. showed that LMWHs did not stimulate osteoclast activity at concentrations that fall within their therapeutic range. They concluded that size and sulfation are major determinants of heparin’s ability to promote bone resorption and that the risk of heparin-induced osteoporosis may be reduced by the use of LMWH preparations: More than 50-fold higher LMWH concentrations were required to obtain an effect equivalent to UFH [27]. In vitro studies have demonstrated that LMWHs produce a significant dose-dependent reduction in osteoblast differentiation and proliferation, as measured by the osteoblast-specific markers osteocalcin and alkaline phosphatase [28]. Heparin fragments with a mean molecular weight of less than 3000 Dalton affect neither osteoblast differentiation nor mineralisation [29]. All these studies suggest that the effects of heparin and its derivatives on osteoblasts are molecular-weight-dependent [28,29,30].

In 2007 Irie et al. demonstrated that heparin enhances osteoclastic bone resorption by inhibiting osteoprotegerin (OPG) activity in vitro. Heparin binds specifically to OPG and prevents its interaction with RANKL on the osteoblastic membrane, so promoting RANK–RANKL interaction and activation of osteoclasts. LMWHs (approximately 4000–6000 Dalton) cause less osteoporosis than standard-size UFHs (approximately 7000–25,000 Dalton), suggesting that UFHs are more inhibitory to OPG than are LMWHs because they are bulkier and sterically hinder OPG–RANKL interaction [31]. Also, Vik et al. investigated the effect of UFH and LMWH dalteparin on plasma levels of OPG in 20 volunteer students; they concluded that UFH causes a more pronounced vascular mobilisation of OPG than does dalteparin, indicating that UFH has a higher affinity for OPG than does LMWH [32].

### 3.2. Effect on Bone Mineral Density (BMD)

The effects of UFH on bone have been mainly studied in the pregnant population. Three clinical studies, including a total of 237 pregnant women undergoing long-term UFH (>6 months), found a significant reduction in BMD [21,22,33]. Barbour et al. evaluated the subclinical occurrence of heparin-induced osteoporosis in a prospective cohort of 14 pregnant women requiring heparin therapy by means of bone densitometry. Five of the 14 cases (36%) had a ≥10% decrease from the baseline proximal femur measurements to immediate postpartum values versus none of the 14 matched controls (*p* = 0.04); this difference continued to be statistically significant six months post-partum (*p* = 0.03) [21]. In the same manner, Douketis et al. [33] found a 7% reduction in BMD when treating pregnant women with long-term heparin therapy, while Dahlman et al. [22] found a 5% reduction in BMD.

The effects of LMWHs on bone density are less researched and more controversial. Most of the studies that examined the effects of LMWHs on bone have also used pregnant women as their patient population. The use of LMWHs has been found to cause less important decreases in BMD. The initial studies showed that more significant reductions were observed in patients receiving enoxaparin for one year or more [34,35,36,37].

However, two reviews reported in depth the effects of long-term LMWH prophylaxis on bone in the pregnant population. These studies have shown that their administration for at least three months was associated with bone loss and fractures [38,39].

It is noteworthy that some clinical studies attributed the occurrence of osteoporosis in this population to pre-existing adverse conditions of the patients: immobilisation, breast feeding and/or pregnancy itself [40]. Hirsch [41] and Schulman [20] have suggested that reduction in BMD caused by the prophylactic use of UFHs and LMWHs were similar to the bone loss verified in physiological pregnancy.

In contrast, Wawrzynska et al. [42] compared the effects of LMWHs, nadroparin and enoxaparin, to those of the VKA acenocoumarol. A decrease in BMD was observed in all three groups. However, decreases in BMD were more prominent in the group of patients receiving enoxaparin for one year or more than in the nadroparin- and acenocoumarol-treated groups. Collectively, while these studies suggest that LMWHs are less deleterious to bone, their long-term use may not be without some risk [43].

In non-pregnant adult populations, the effects of long-term use of LMWHs and UFHs on BMD are unclear. Bernis [44] and Lai [45] conducted two clinical trials in haemodialysis patients and found insignificant changes in mean BMD, with more important changes in patients using UFH compared to LMWH. Two other trials evidenced the absence of bone loss in non-pregnant populations [43,46]. In contrast to these clinical trials, several cohort studies showed a significant reduction in BMD with long-term LMWH treatment [47,48].

### 3.3. Effects on Fractures

The possible involvement of heparins as a risk factor for fractures is highly controversial. Several studies have calculated the risk of fracture caused by long-term heparin treatment. Pettila [49] reported a vertebral fracture rate of 3.6% in pregnant populations that received long-term UFH, while no fractures were found in long-term LMWH populations. Similarly, Dahlman [50] reported only 2.2% fracture rate while Monreal [51] found a vertebral fracture rate of 15% in patients that received long-term UFH compared to only 2.5% rate for long-term LMWH. These studies confirm the prominent role played by UFH in changes in bone metabolism and a lower risk of fractures in patients with long-term LMWH treatment.

A recent meta-analysis (2016) conducted by Gajic-Veljanoski determined the effects of LMWH therapy of at least three month’s duration on fractures and BMD in non-pregnant adult populations. LMWH for three to six months may not increase the risk of fractures, but longer exposure of up to 24 months may adversely affect BMD [52].

## 4. Oral Anticoagulants (VKAs)

Oral anticoagulants have been associated with an increased risk of osteoporosis when used long-term or the treatment and prevention of thromboembolic diseases [6]. According to different clinical indications, these medicines could be prescribed for several months but, in many cases, they can be taken for many years or lifelong.

### 4.1. Effects on Bone Metabolism

These drugs are vitamin K antagonists and produce their anticoagulant effect by interfering with and inhibiting vitamin K epoxide reductase. Thus, they modulate the carboxylation range of glutamic acid residues not only in coagulation factors II, VII, IX, and X, but also in osteocalcin, a bone-specific protein. The osteocalcin is synthesized and subsequently gamma-carboxylated by the vitamin K-dependent gamma-glutamyl carboxylase localised in osteoblasts. The resulting carboxylated osteocalcin (Gla-Oc) binds to the hydroxyapatite of bone and accumulates in the bone matrix, while the incompletely gamma-carboxylated form (Glu-Oc) has a low affinity for bone matrix and is released into the blood and acts on pancreatic B cells to enhance insulin secretion and thus influence glucose metabolism [53].

Another possible indirect mechanism of bone deterioration is related to dietary restrictions frequently adopted in patients using VKAs [54,55,56,57,58,59].

### 4.2. Effect on BMD

In a prospective observational study conducted in 6201 postmenopausal women, the authors showed that warfarin users (*n* = 149) compared with warfarin nonusers (*n* = 6052), more frequently had poor health, involuntary weight loss, non-thiazide diuretic use and frailty, but had similar BMD at the hip (difference, 1.6% (95% CI–0.7% to 4.1%)) and heel (difference, 2.1% (95% CI–1.6% to 5.6%)) [60].

Avgeri et al. evaluated bone turnover markers and BMD in 23 children undergoing long-term therapy with VKAs compared to 25 control children. The authors observed that patients using VKAs showed increased levels of Glu-Oc, parathormone (PTH) and bone resorption markers, and lower levels of bone formation markers and 25-OH vitamin D. Of the patients, 52% were osteopenic. Statistical data analysis showed that long-term therapy with VKAs was an independent predictor of changes in bone turnover markers, suggesting that long-term therapy could cause osteopenia in children and a consequent risk of developing osteoporosis [61].

In another case–control study, performed in 70 patients with rheumatic valvular heart disease subjected to mechanical valve replacement and on long-term warfarin compared to 103 randomly selected matched controls, a detrimental long-term effect of VKAs on BMD was demonstrated. The authors, who observed a marked reduction in BMD of lumbar spine in patients compared to controls, suggested an evaluation over time of the BMD and the prophylactic use of calcium and vitamin D supplements [62].

In a systematic review of nine original cross-sectional studies on the effect of long-term exposure to any oral anticoagulant on bone density in adults, BMD was significantly decreased among exposed subjects only in the ultradistal radius, but not in other sites (distal radius, lumbar spine, femoral neck or femoral trochanter). This suggests that long-term oral anticoagulation use might be associated with no more than a modest increase in osteoporotic fracture risk, but this should be verified in future longitudinal studies [63].

In conclusion, even if data are contrasting, VKAs seem to lead to a reduction in BMD, especially in patients undergoing long-term treatment with this class of drugs. However, it should be specified that the critical duration of exposure to VKAs that may result in an increased risk of fractures is not yet known, as several studies have not estimated a sufficiently long time of exposure to VKAs [64]. The presence of vitamin K in the diet should be allowed in patients using VKAs in relation to the patient’s comorbidities and eating habits; to prevent osteoporosis an adequate presence of protein, calcium and vitamin D should, in addition, be guaranteed, the latter also increased by a specific supplementation [65].

### 4.3. Effects on Fractures

Jamal et al., in a prospective observational study, evaluated the risk of fractures in 6201 postmenopausal women during 3.5 years of follow-up and showed that warfarin users and nonusers showed similar rates of bone loss (1.1% and 0.8%; *p* = 0.18) and fractures (relative hazard 1.0; CI 0.60 to 1.71) [60].

In a population-based retrospective cohort study, 572 women 35 years or older at their first lifetime venous thromboembolism event were followed up for fractures, between 1966 and 1990. Risk was assessed by comparing new fractures with the number expected from sex- and age-specific fracture incidence rates for the general population. Exposure to oral anticoagulation was associated with an increased risk of vertebral and rib fractures. Oral anticoagulation for 12 months or more was an independent predictor of vertebral fractures (*p* = 0.009) and rib fractures (*p* = 0.02), but not other fractures [64].

In another retrospective cohort study of 4461 patients undergoing long-term warfarin therapy (≥1 year), the authors observed a significant association between osteoporotic fracture and warfarin use in men (OR 1.63; 95% CI 1.26–2.10) but not in women (OR 1.05; 95% CI 0.88–1.26). Moreover, in patients (of both sexes) prescribed warfarin for less than 1 year, the risk of osteoporotic fracture was not significantly increased (OR 1.03) [66].

A very recent meta-analysis study found that the patients receiving VKAs who had a higher risk of fractures were those over 65 years of age and women, but the strength of this association was weak and clinically insignificant, as other factors came into play [67].

Ultimately one might conclude that there is no definitive and strong association between the use of VKAs and fractures for several reasons. Some studies consider only clinical fractures, but especially vertebral fractures are subclinical and may not be reported by patients. Many studies do not evaluate adherence to VKAs. Finally, there is always the risk of selection bias (many populations studied may have other risk factors for fractures) [68,69].

## 5. Direct Oral Anticoagulants

The possible effects that direct oral anticoagulants (DOACs) can induce in bone metabolism has so far been poorly studied. The few available studies are uneven and are predominantly cohort studies. The general idea that emerges is that their use has a mild effect on any changes in bone metabolism [70,71,72,73,74,75,76,77,78] and there is no significant clinical consequence to date to the extent that they cause a reduction in bone mass or secondary osteoporosis fractures. Comparison with VKAs shows a lower risk of osteoporosis and fractures [79,80].

### 5.1. Effects on Bone Metabolism

In 2011, Gigi et al. investigated the effects on bone biology of DOACs using an in vitro cell culture model from the human female osteoblastic cell line SaOS-2 [70,71]. Cells at subconfluence were treated for 24 hours with different concentrations of rivaroxaban and analysed for DNA synthesis and creatine kinase and alkaline phosphatase-specific activities. Treatment was continued for 21 days to analyse mineralisation. Rivaroxaban inhibited dose-dependently up to 60% DNA synthesis by the cells. Creatine kinase-specific activity was also inhibited dose-dependently to a similar extent, whereas alkaline phosphatase-specific activity was inhibited up to 30%. Osteoblastic mineralisation was unaffected, indicating that rivaroxaban inhibits the first stage of bone formation but does not exert effects in later stages (bone mineralisation), thus producing a transient inhibition of bone formation [70]. The same authors investigated the effects of rivaroxaban on the response of cell line SaOS-2 to osteoblast-modulating hormones. The results suggested that rivaroxaban may exert inhibitory effects not only in the early stages of bone formation, but may also affect the stimulatory effects of bone modulating hormones with mechanisms as yet unclear [71].

Winkler et al. investigated the effect of a direct thrombin inhibitor (melagatran) compared to dalteparin and UFH on human osteoblasts. Melagatran, dalteparin and UFH were added to primary osteoblast cultures in their therapeutic range and in two decimal powers below and above. Cell number, protein synthesis, mitochondrial and alkaline phosphatase activity and collagen type I synthesis were evaluated. Melagatran showed fewer inhibitory in vitro effects on human osteoblasts than did dalteparin or UFH; the latter showed the most pronounced influence on cellular metabolism [72].

In 2013 Morishima et al. determined the effects of warfarin and edoxaban on the serum concentration of total γ-carboxylated osteocalcin (Gla-Oc) and under-carboxylated osteocalcin (Glu-Oc) in rats. The animals received orally administered warfarin or edoxaban, and 24 hours later serum and plasma samples were obtained for osteocalcin and prothrombin time (PT) measurements. Warfarin at 1 mg/kg markedly increased serum levels of Glu-Oc and slightly increased those of total osteocalcin compared to control rats. Serum Gla-Oc significantly decreased after warfarin ingestion. Edoxaban at 1 mg/kg (antithrombotic dose) and 54 mg/kg (a dose which prolonged PT 2.25-fold) had no effects on levels of total osteocalcin, Glu-Oc and Gla-Oc. This study suggests that in rats, in contrast to warfarin, edoxaban, at doses higher than those needed for its antithrombotic activity, has no effects on the production of Gla-Oc and thus, may have a lower risk of adverse effects on bone health [73]. In contrast, Gandhi found that rivaroxaban treatment may negatively affect bone through a reduction in osteoblast function. This reduction was associated with a reduction in the mRNA expression of the bone marker osteocalcin, the transcription factor Runx2, and the osteogenic factor BMP-2 [74].

Pilge et al. evaluated the effects of rivaroxaban and enoxaparin on the proliferation, mRNA and surface receptor expression, as well as differentiation capacity, of primary human mesenchymal stromal cells (hMSCs) during their osteogenic differentiation. Enoxaparin, but not rivaroxaban treatment, significantly increased hMSC proliferation during the first week of osteogenic differentiation while suppressing osteogenic marker genes, surface receptor expression and calcification, indicating that rivaroxaban seems to be superior to enoxaparin in the early stages of bone healing in vitro [75].

Namba et al. evaluated the effects of changing from warfarin to rivaroxaban on bone mineral metabolism, vascular calcification, and vascular endothelial dysfunction. The study reported 21 consecutive patients with persistent or chronic AF, who were treated with warfarin for at least 12 months. Warfarin administration was changed to rivaroxaban (10 or 15 mg/day) in all patients. Switching to rivaroxaban from warfarin in patients with AF was associated with an increase in bone formation markers and a decrease of bone resorption markers [77].

### 5.2. Effect on BMD

No data resulted about the effect of DOACs on BMD.

### 5.3. Effects on Fractures

In 2017 Lau et al. investigated the risk of osteoporotic fractures with dabigatran versus warfarin in patients with nonvalvular atrial fibrillation (NVAF). Among adults with NVAF receiving anticoagulation, the use of dabigatran compared to warfarin was associated with a lower risk of osteoporotic fracture. Additional studies, perhaps including randomised clinical trials, may be warranted to further understand the relationship between use of dabigatran versus warfarin and the risk of fracture [79].

Treceño-Lobato et al. included a total of 334 patients in a study with the aim to evaluate the adverse drug reaction (ADR) incidence rate for classic compared to new anticoagulants. The most significant result was the osteoporosis risk associated with the consumption of VKAs, with 11 cases recorded in the cohort of these patients versus none in the cohort of patients treated with DOACs, supporting the appropriateness of using DOACs in patients at risk of osteoporotic fractures [80].

## 6. Conclusions

Data of literature evidence a heterogeneous effect of anticoagulant on bone. Table 2 summarises the effects on bone metabolism, BMD and fractures of the different available anticoagulants. Data supporting a detrimental effect of heparin are sufficiently clear. LMWHs seem to be safer than heparin. Although VKAs have a significant impact on bone metabolism and, in particular, on osteocalcin, data on BMD and fractures are contrasting. To date, the new DOACs are safe for bone health (Figure 1). This evidence must be taken into consideration by medical personnel. In particular, the prolonged use of heparin and VKAs must induce an alert behaviour and put in place some prophylaxis measures, such as calcium and vitamin D supplementation. In the case of a history of fractures, or in the presence of other comorbidities or low BMD values, it is necessary to evaluate a therapy with anti-osteoporosis drugs.

## Figures and Tables

**Figure 1 ijms-20-05275-f001:**
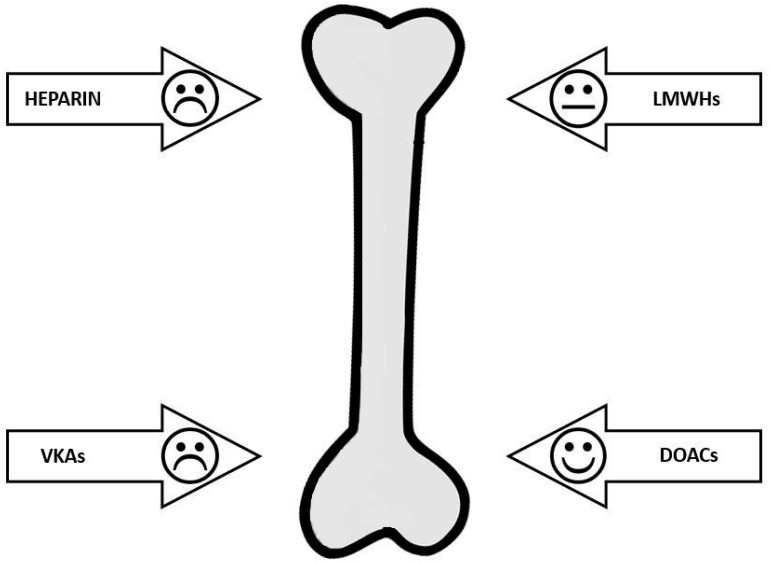
Anticoagulants’ safety on bone. LMWHs (Low molecular weight heparins), VKAs (Vitamin K antagonists), DOACs (Direct-acting oral anticoagulants).

**Table 1 ijms-20-05275-t001:** Vitamin K antagonists therapy: indications.

VKA Therapy: Indications
Venous thromboembolism prophylaxis [6]
Pulmonary embolism and deep vein thrombosis therapy [9]
Thromboembolic prophylaxis in patients with dilated cardiomyopathyc [14]
Thromboembolic prophylaxis in patients with biologic heart valve replacement [6]
Thromboembolic prophylaxis in patients with myocardial infarction (primary, secondary prevention) [15]
Thromboembolic prophylaxis in patients with aortic valvular heart disease [16]
Thromboembolic prophylaxis in patients with mechanic heart valve replacement [6]
Thromboembolic prophylaxis in patients with anti-phospholipid syndrome [9]
Thromboembolic prophylaxis in patients with atrial fibrillation [6]

**Table 2 ijms-20-05275-t002:** Summary of the effects of anticoagulants on bone metabolism, bone mineral density (BMD), and fractures.

Drug Class	Effect on Bone Metabolism	Effect on BMD	Effect on Fractures
Heparin	High	High	High
Low molecular weight heparins	Low	Uncertain	Uncertain
Vitamin K antagonists	High	Low	Low
Direct-acting oral anticoagulants	None	N/A	None

N/A: data not available.

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
