# Peer review of "Anticoagulants and Osteoporosis"

_ijms, 2019, doi:10.3390/ijms20215275_

Round 1

Reviewer 1 Report

This review article discusses the effects of anticoagulants on bone metabolism, bone mineral density and on fragility fractures. This manuscript is not quite interest for me because authors just summarized the previous studies of different groups of anticoagulants, Heparin, LMWHS, oral anti-vitamin K agents, and direct oral anticoagulants in bone health. I don’t know well what is the key points that authors focused on and want to say to the readers.

There are major concerns that should be addressed.

For the better understanding the review, I recommend authors to add appropriate diagrams describing the text. Authors need to add a detailed summary and the aims of this review in Abstract and Introduction sections, respectively. In Table 1, authors should appropriate references. In Table 2, all abbreviations should be changed to the full descriptions. In Conclusion section, authors should discuss how they could reach that conclusions in detail and should give their opinion.   In keywords, the word ‘LMWHs’ is better to be changed to ‘low molecular weight heparins (LMWHs)’

Author Response

For the better understanding the review, I recommend authors to add appropriate diagrams describing the text. Authors need to add a detailed summary and the aims of this review in Abstract and Introduction sections, respectively. In Table 1, authors should appropriate references. In Table 2, all abbreviations should be changed to the full descriptions. In Conclusion section, authors should discuss how they could reach that conclusions in detail and should give their opinion.   In keywords, the word ‘LMWHs’ is better to be changed to ‘low molecular weight heparins (LMWHs)’ 

Response: Thank you for your useful suggestions.

We added a figure at the end of the manuscript that summarizes the anticoagulants’ safety on bone. Abstract and introduction were completed according to your suggestions. Table 1 shows now the appropriate references. In Table 2, all abbreviations have been changed to the full descriptions. Conclusion section was rewritten. In keywords, the acronym LMWHs has been changed to “low molecular weight heparins”.

Reviewer 2 Report

This is a comprehensive and accessible review of evidence currently on the effects of anticoagulants on bone metabolism, BMD and fractures. The manuscript is well written and very timely given the extensive use of anticoagulant drugs in prophylactic treatment.

Author Response

This is a comprehensive and accessible review of evidence currently on the effects of anticoagulants on bone metabolism, BMD and fractures. The manuscript is well written and very timely given the extensive use of anticoagulant drugs in prophylactic treatment. Publication is recommended.

Response: Thank you for your encouraging judgment.

Round 2

Reviewer 1 Report

No comment.